# Inhibition of Caries around Restoration by Ion-Releasing Restorative Materials: An In Vitro Optical Coherence Tomography and Micro-Computed Tomography Evaluation

**DOI:** 10.3390/ma16165558

**Published:** 2023-08-10

**Authors:** Eman H. Albelasy, Ruoqiong Chen, Alex Fok, Marmar Montasser, Hamdi H. Hamama, Salah H. Mahmoud, Tamer Abdelrehim, Hooi Pin Chew

**Affiliations:** 1Conservative Dentistry Department, Faculty of Dentistry, Mansoura University, Algomhoria Street, Mansoura 35516, Egypt; emanhamza92@mans.edu.eg (E.H.A.); marmar1659@gmail.com (M.M.); hamdy_hosny@mans.edu.eg (H.H.H.); salahmahmoud2010@mans.edu.eg (S.H.M.); 2Minnesota Dental Research Center for Biomaterials and Biomechanics, School of Dentistry, University of Minnesota, Minneapolis, MN 55455, USA; alexfok@umn.edu (A.F.);; 3Department of Diagnostics and Biological Sciences, School of Dentistry, University of Minnesota, Minneapolis, MN 55455, USA; 4Faculty of Dentistry, New-Mansoura University, New-Mansoura 35712, Egypt; 5Conservative Dentistry Department, Faculty of Dentistry, Horus University, New-Dumyat 34517, Egypt

**Keywords:** dental restoration, optical coherence tomography, dental caries, biofilms

## Abstract

The objective of this study was firstly to assess the demineralization inhibitory effect of ion-releasing restorations in enamel adjacent to restoration using a biofilm caries model and secondly to compare the effect to that in a chemical caries model. Fifty-six bovine incisors were filled with either Surefil one (SuO), Cention N (CN) (both ion-releasing materials), Ketac-Molar (GIC) or Powerfill resin composite (RC). The restored teeth were then randomly divided into 2 groups according to the used caries model (biofilm or chemical caries model). The micro-computed tomography (MicroCt) and optical coherence tomography (OCT) outcome measures used to evaluate demineralization inhibition effects were lesion depth, *LD* and increase in OCT integrated reflectivity, Δ*IR*, at five different depths. It was observed that all outcome measures of CN were statistically the same as those of GIC and conversely with those of RC. This was also the case for SuO except for *LD*, which was statistically the same as RC. When comparing the two caries models, *LD* of the biofilm model was statistically deeper (*p* < 0.05) than the chemical model for all four materials. In conclusion, CN and SuO have similar demineralization inhibitory effects as GIC, and the biofilm caries model is more discriminatory in differentiating demineralization inhibitory effects of ion-releasing restorative material.

## 1. Introduction

With resin composite approaching 50 years of clinical use, several developmental modifications have been made to address its clinical performance deficiencies [1]. Despite advancements in resin composite formulations having broadened their clinical indications [2,3], their longevity is frequently found to be limited by the development of carious lesions at and near the tooth-restoration interface [2,4]. These carious lesions have been widely known as secondary caries or, more recently, caries around restoration or CARS [5]. While the direct cause and effect of polymerization shrinkage stress on CARS has not been unequivocally proven [6], it is evident that shrinkage stress causes gap formation, enamel cracks and marginal leakage [7]. Such findings show that it is beneficial for composite restorations to have antimicrobial and/or remineralizing properties either to inhibit or reduce the rate of secondary caries formation.

Despite the terminology debate, materials claiming to have bioactive properties have become a topic of interest for clinicians and manufacturers. While some have defined bioactive materials as materials that are able to form apatite-like compounds on their surfaces when immersed in a physiological-like solution [8], others disagree and argue that for a material to be called bioactive, it has to interact with living tissues without the essential modulation of a natural biochemical process and without resorting to non-natural substances [9].

Glass ionomers (GIC) and resin-modified glass ionomers (RMGIC) are examples of materials that are capable of forming a bioactive interface with biological tissues. These materials have the capabilities to release and be recharged with fluoride ions, thus promoting remineralization [10]. Furthermore, ionic bonds form between the carboxylate groups on the polyacid molecules and calcium ions in the tooth surface [11]. Stemming from the need to combine the time-efficient benefits of glass ionomer application and its fluoride-release ability along with the durability of resin composite restorations, several new bulk-fill ion-releasing resin composites have been introduced. One of the commercially available ion-releasing resin composites [10], Cention N™ (CN), (Ivoclar Vivadent, Schaan, Liechtenstein), contains reactive glasses that were shown to release Na^+^, Ca^2+^ and F^−^ ions [12] and, in an in vitro study, increased the pH of saliva and formed apatite when immersed in artificial saliva [13]. According to the manufacturer, CN contains three inorganic glasses: a conventional inert barium-aluminosilicate glass, an ionomer glass based on a calcium barium alumino-fluorosilicate (FAS) and a basic calcium fluorosilicate glass referred to as an “alkasite” filler [12,13]. An energy dispersive X-ray composition analysis (EDX) of CN revealed a relatively large amount of calcium [14], which is claimed by the manufacturer to impart remineralization potential. Another development in the same perspective is a hybrid, dual-polymerized bulk-fill resin composite, Surefil one™ (SuO) (Dentsply-Sirona, Konstanz, Germany). In addition to being self-adhesive, it is claimed by the manufacturer to release fluoride.

Due to their relative novelty, these new materials are under intense investigation both in vitro and in vivo, especially regarding their ion release and its inhibitory effect on demineralization and CARS. While the caries inhibitory effect of glass ionomers has been established in many laboratory studies [15,16], the anti-caries properties of these resin-based materials are still scarcely reported [17,18].

Many artificial caries models have been developed to investigate the de- and remineralization processes of tooth structure, and they can be divided into two main categories, i.e., chemical or biological. Chemical models were one of the first to be used to simulate carious lesions [15] either by the static exposure of dental hard tissues to a demineralizing solution or dynamic pH cycling between acidic demineralizing solutions and neutral remineralizing solutions in order to mimic the intraoral conditions [5]. Other models are based on the direct exposure of restored hard tissues to low pH generated by a biofilm either statically in well plates or dynamically in a bioreactor [16]. While biological models are more complex than chemical ones, they provide valuable information about the nature of the carious lesions, as they resemble the oral environment in terms of the presence of bacteria, the fluctuating pH and the shear-flow, which are ultimately responsible for many of the morphological features and behavior of the biofilms themselves [19]. Biofilm models can be mono-species, multi-species or a microcosm biofilm model [15]. Biofilm-based in vitro studies are less expensive than clinical trials and more clinically relevant than chemical models because the biofilm structures and processes that mediate tooth demineralization and material degradation are replicated [20]. Nevertheless, it is important to note that ideally, in vitro results should be calibrated against clinical data. More studies that compare in vitro and in vivo data should be conducted to relate in vitro results to the clinical scenario [15].

Previous research [16] that compared biofilm and chemical models for secondary caries formation around restorations indicated comparable outcomes in evaluating the impact of restorative materials, despite variability in lesion severity. However, to replicate the results of biofilm models, chemical models must be modified by adjusting the type of acid used, the duration and the pH levels [21].

Optical coherence tomography (OCT) is a new diagnostic technique for imaging internal biological structures in cross-sections. OCT aids in the visualization of differences in tissue optical properties, including the effects of optical absorption and scattering. It is an interferometric technique that employs near-infrared light waves that reflect off the internal microstructure in a manner analogous to an ultrasonic pulse echo in theory [22]. Previous research measured the birefringence of dentin and enamel and proposed that the enamel rods functioned as waveguides [23] because the scattering properties of restorative materials and dental hard tissues differ significantly; OCT can be used to detect secondary caries [24].

The objectives of this in vitro study are to:1.Compare the demineralization inhibitory effect of two ion-releasing restorative materials to that of a conventional resin composite (negative control) and a conventional glass ionomer material (positive control).2.Evaluate the influence of the type of artificial caries model on the demineralization inhibitory effect of the four materials mentioned above.

Hence, our null hypotheses are:1.There are no significant differences in the demineralization inhibitory effect of the ion-releasing restorations from that of resin composite.2.Artificial caries models have no effect on the demineralization inhibitory effect of the ion-releasing restorations.

## 2. Materials and Methods

### 2.1. Restorative Materials

Four restorative materials were used in this study: Surefil one (Dentsply, Sirona), Cention N (Ivoclar Vivadent, Schaan, Lichtenstein), Ketac^TM^ Molar Aplicap (3M Espe, St. Paul, MN, USA) and a bulk fill resin composite, Tetric PowerFill (Ivoclar Vivadent, Schaan, Lichtenstein). Materials’ classifications and lot numbers are illustrated in Table 1.

### 2.2. Sample Size Calculation

Sample size calculation was based on a previous study with a similar study design that showed a statistically significant difference (*p* < 0.050) between the ion-releasing materials and the control group [25]. A total of 56 teeth achieved 80% power and 0.05 type 1 error.

### 2.3. Sample Preparation

Sample preparation and study flow are illustrated in Figure 1. Fifty-six bovine incisors were collected and cleaned from any calculus deposits using a manual scaler (Zeffiro Jacquette Scaler, Lascod, Sesto Fiorentino, Italy). The collected teeth were checked under a stereomicroscope to exclude teeth with visible cracks. They were also inspected for any pre-existing demineralization using OCT (IVS-2000, Santec, Komaki, Japan). Any teeth with demineralization, observed visually by an increase in backscattered intensity on the false color map “*Upper 45*” on the Santec *Innervision* software (Komaki, Japan), were discarded. The teeth were then stored in 0.1% thymol at 4 °C until use. The storage solution was changed every 2 weeks. All collected teeth were used within 3 months of storage. The teeth were mounted in acrylic approximately 2 mm apical to the cemento-enamel junction using a specially designed jig device for the standardization of the position and angulation of each tooth inside the ring during mounting.

All cavity preparation and restoration procedures were performed by a single trained operator (E.A). Cuboidal cavities with widths of 3 × 3 mm^2^ and a depth of 2 mm were prepared at the cement-enamel junction with half of the cavity in enamel and the other half in root dentine. The preparation was performed using a diamond bur (SF15, SS-White Dental, Lakewood, NJ, USA) under copious water cooling using a high-speed handpiece (LS 22K, Brasseler, Savannah, GA, USA). The prepared cavity was finished with a yellow-coded diamond (FO21E, SS-White Dental, Lakewood, NJ, USA). No bevels were performed at the cavosurfaces, and the burs were discarded every 5 preparations. After that, the samples were coated with an acid-resistant nail varnish (Revlon, New York, NY, USA), leaving 2 mm of exposed enamel or dentin around the cavity preparations. 

The teeth with cavity preparations were then randomly assigned to 4 groups according to the type of restorative material: (Surefil one^TM^, Dentsply, Sirona), Cention N (Ivoclar Vivadent, Schaan, Lichtenstein), Ketac^TM^ Molar Aplicap (3M Espe, St. Paul, MN, USA) and bulk fill resin composite, (Tetric Powerfill, Ivoclar Vivadent, Schaan, Lichtenstein). The randomization was performed using manual block randomization with a balanced block size of 8. Each restorative material was given a number from 1 to 4, written on a piece of paper and then randomly assigned to each block by shuffling the paper. 

### 2.4. Restoration Placement

After the cavity preparation procedures, the cavities were dried with oil-free air syringe without desiccation.

#### 2.4.1. Surefil One

According to the manufacturer’s instructions, the activated capsules were mixed for 10 s in a capsule mixer (Roto-mix, 3M Espe, St. Paul, MN, USA). Using a capsule extruder, the material was dispensed directly into the cavity from the capsule tip, the cavity was filled in bulk and the tip was gradually withdrawn and then light cured for 30 s with a light curing unit (Elipar™ S10 curing light, 3M ESPE, St. Paul, MN, USA). A transparent matrix band was used to increase adaptation to the cavity margins. Any gross excess material was removed using a surgical scalpel blade no.12 (Devemed GmbH, Tuttlingen, Germany). The restored cavity was then polished using flexible aluminum oxide discs (Sof-lex^TM^3M ESPE, St. Paul, MN, USA).

#### 2.4.2. Cention N

Cention N was applied in conjunction with a universal adhesive (Adhese Universal, Ivoclar Vivadent, Schaan, Lichtenstein) applied in selective-etch mode. Enamel margins were etched with 37% phosphoric acid (N-etch, Ivoclar Vivadent, Schaan, Lichtenstein) for 20 s, rinsed for 20 s and then gently air-dried. One coat of the universal adhesive (Adhese Universal, Ivoclar Vivadent, Schaan, Lichtenstein) was applied with agitation for 20 s and light cured using the light curing unit (Elipar™ S10, 3M ESPE, St. Paul, MN, USA) for 10 s. With a mixing ratio of 1:1, Cention N was mixed on a paper pad with a plastic spatula until a homogenous mix was reached. The mix was then placed in bulk in the cavities and condensed using a metallic condenser. A transparent matrix was placed over the restoration to improve adaptation. The material was left to chemically set. 

#### 2.4.3. Conventional GIC

Conventional GIC (Ketac^TM^ molar AplicapTm, 3M Espe, St. Paul, MN, USA) was activated in a capsule activator by holding down the lever for 2 s. The activated capsule was then mixed in the capsule mixer (Roto-Mix, 3M Espe, St. Paul, MN, USA) for 8 s. The capsule was then placed in a capsule extruder and dispensed into the cavity. The cavity was filled in bulk, and the tip was gradually withdrawn. A ball-burnisher was used to adapt it to the cavity margins and left to chemically set.

#### 2.4.4. Powerfill Resin Composite

Powerfill bulk-fill composite was applied in conjunction with a universal adhesive applied in selective-etch mode. Enamel margins were etched with 37% phosphoric acid, rinsed for 20 s and then gently dried. One coat of a universal adhesive (Adhese Universal, Ivoclar Vivadent, Schaan, Lichtenstein) was applied with agitation for 20 s and light cured for 10 s. Powerfil composite was applied in bulk and cured for 10 s.

The restored teeth were stored in distilled water in an incubator at 37 °C (HeraTherm, Thermo Scientific, Waltham, MA, USA) for 24 h before finishing and polishing procedures. The restored teeth were polished using aluminum oxide discs (Sof-Lex^TM^, 3M Espe, St. Paul, MN, USA). The teeth were cleaned from any debris using an ultrasonic cleaner (Sonicator Instrument Corporation, Copiague, NY, USA). All preparations were performed by a single operator.

### 2.5. Artificial Secondary Caries Models

Simple randomization was then performed in each restoration group to assign the samples to either the pH cycling or the biofilm artificial caries model, resulting in 8 groups (n = 7). 

#### 2.5.1. pH Cycling Model

The demineralization and remineralization solutions were prepared according to a modified Featherstone pH cycling model for bovine incisors [26]. The demineralization stage (6 h) uses an acid buffer containing 2 mM Ca (NO_3_)_2_·4H_2_O, 2 mM (KH_2_PO_4_) and 75 mM acetate at pH 4.5. The remineralization solution (18 h) contains calcium and phosphate at a known degree of saturation (1.5 mmol/L Ca (NO_3_)2·4H_2_O, 0.9 mmol/L KH_2_PO_4_). To mimic the remineralizing properties of saliva, 130 mM KCl was used (to provide background ionic strength) along with 20 mL Na cacodylate buffer at pH 7. The demineralizing solution was changed twice a week, and the remineralizing solution was changed 3 times a week. Samples were placed in sterile vials with 20 mL of remineralizing solution per sample and 40 mL of demineralizing solution per sample. Before changing the solution, each tooth was cleaned with deionized water. The samples were kept in an incubator at 37 °C on an orbital shaker for 9 days.

#### 2.5.2. Single-Species Dynamic Biofilm Model 

##### The Center for Disease Control (CDC) Reactor

The CDC reactor (BioSurface Technologies, Bozeman, MT, USA) is a 1-L lidded vessel with an influx port at the top and an effluent port at a height of 400 mL. During the challenge, a magnetically driven vaned stir bar kept the media constantly mixed [27]. A trial run of the reactor showing its components is illustrated in Figure 2. The lid allowed for the incorporation of a pH electrode, temperature probe and six rods to accommodate the samples. For this experiment, a new sample holder was created that fits 6 mounted bovine incisors per cycle. Before placing the samples in the reactor, they were immersed in 75% ethanol for 1 min. The CDC reactor was placed in the fume-hood during the experiment. Sample manipulation was performed in a bio-safety cabinet (Class II BSC Airstream, ESCO, Singapore).

##### Preparation of Bacterial Suspensions 

Streaks of *Streptococcus mutans* (ATCC700610) were grown on a mitis-salivarius sucrose bacitracin agar and incubated in a 5% CO_2_ incubator for 48 h at 37 °C. A few colonies were inoculated in a 5 mL of brain heart infusion broth (BHI), and incubated in a 5% CO_2_ incubator for 24 h at 37 °C. The media was sterilized prior to the start of the experiment in an autoclave (LV 250 laboratory steam sterilizer, STERIS, Basingstoke, United Kingdom) at a temperature of 121 °C, and a pressure of 15 psi.

Serial dilution of the culture was performed (OD = 1.0) in BHI-sucrose media at 1:100. Three hundred-and fifty ml of diluted culture were added into the CDC reactor with 0.1% sucrose. The temperature was set at 37 °C, and the pH reading was recorded at 15-min intervals using a pH meter (AB15 Plus, Accumet Basic, Fisher Scientific, Hampton, NH, USA). The samples were incubated in the CDC reactor at 37 °C under 89 rpm shear rate but with no fresh media flow for the first 24 h. After 24 h, the reactor was connected to the nutrient carboy, and fresh BHI broth with 0.1% sucrose was pumped at a flow rate of 0.1 mL/min for 5 days.

The flow rate was determined after a pilot study where the pH was found to be around 4.2–4.5 at a flow rate of 0.1 mL/min. The experiment was terminated after 6 days. The reactor components were washed with bleach diluted with water (1:10) and then autoclaved. After the demineralization challenge and before scanning, the samples were placed in 75% ethanol to avoid contamination of the imaging equipment.

Following the termination of the experiment, the restored samples were kept in 4 °C until the scanning procedure.

### 2.6. Swept-Source Optical Coherence Tomography (SS-OCT) Scanning and Data Processing

All samples were scanned with a SS-OCT system (IVS-2000, Santec, Komaki, Japan) before and after the demineralization challenge. The scans taken before demineralization served as baseline for later comparison and computation of outcome measures. The wavelength of the SS-OCT used in this study ranged from 1260 nm to 1360 nm and utilized a high-speed frequency (20 kHz sweep rate) swept source external cavity laser. An area of 8 × 8 mm^2^ was scanned, and that included the restoration in the center and a peripheral 2.5 mm of enamel and dentine on each side of the restoration in the X-Y plane (Figure 3A). The axial and lateral spatial resolution were 4.4 µm (refractive index of 1.63) and 11.4 µm, respectively. The *Innervision* software (Santec, Komaki, Japan) was used to capture and view the scans. The samples were positioned at the same orientation for scanning before and after demineralization, with the aid of a specially designed sample holder that directly fits into the mounted sample. The distance between the laser source and the samples was standardized to be within the focal range of the OCT. 

OCT data processing was performed using custom-written scripts in MATLAB (Mathworks, Natick, MA, USA). Four equally interspersed cross-sections (B-scans) from each 3D scan were selected (Figure 3B,D). OCT integrated reflectivity (*IR*) of the enamel adjacent to the tooth-restoration interface is the target outcome measure of demineralization severity [28] around restoration in this study. As the interval of artificial caries inductions of both the biofilm and chemical models in this study are relatively short, we do not expect wall lesions of caries around restoration to have developed, and hence, the region of interest (ROI) in this study is limited to the surface lesion of CARS, and in the case of this study, a 0.25 mm width of enamel immediately adjacent to tooth-restoration interface is deemed the ROI.

The following are the processing steps applied to the OCT data to compute *IR*. The surfaces of the samples were first determined by means of thresholding. The average reflectance value of background (air in this case) was attained from known area of air immediately above the surface of a reference object. Thereafter, the script locates and designates the first pixel in each depth-resolved reflectance line profile (A-scan) that is higher than the mean reflectance value of air as the surface. After this, the ROIs were manually selected (Figure 4A). In order to derive the *IR* of the ROIs, a mean A-scan for each ROI had to be computed. For this, the determined surface was aligned to the highest axial coordinate of the surface (Figure 4B), and the mean A-scan of the ROIs was computed (Figure 4C). It was previously shown that structural-related reflectivity changes of the top 20 microns in enamel may be masked by surface specular reflectance [29]. Hence, the reflectivity of the top 20 µm was not included in the calculation of *IR* in this study, and the *IR* of the following five depths were computed from the mean A-scan: 30–50 µm (*IR*_50_), 30–100 µm (*IR*_100_), 30–150 µm (*IR*_150_), 30–200 µm (*IR*_200_) and 30–250 µm (*IR*_250_). The increase in *IR* (Δ*IR*) between baseline (T_0_) and post-demineralization (T_1_) of these various depths was the outcome measure used for inter-group demineralization inhibition comparison.

### 2.7. Micro-Computed Tomography (Micro-Ct)

The samples were also scanned using the XT H 225 Micro-Ct machine (XT H 225, Nikon Metrology Inc., Brighton, MI, USA) with the following parameters: 105 kV voltage, 94 μA current, 720 projections and 4 frames/projection. A 0.5 aluminum filter was used to mitigate the beam-hardening phenomenon. The resulting voxel size was 8.7 μm, and the scanning time was 34 min. The mounted samples were placed in a custom-made jig that fit into the mount and stabilized the samples while rotating. Representative samples from each group were scanned prior to the demineralization challenge to set a baseline for sound enamel and dentine for later comparison. Using the same parameters of the baseline scans, all samples were scanned following demineralization.

The images were reconstructed using the CT Pro 3D XT 3.1.11 (Nikon metrology, Inc., Brighton, MI, USA) software, accounting for beam hardening correction with an in-built algorithm. The data were then transferred to VGSTUDIO MAX 3.4 (Volume Graphics Charlotte, NC, USA) software for visualization. Under a standardized viewing magnification factor, five inciso-gingival, equally interspersed cross-sections across the restoration were selected.

Custom-written scripts in MATLAB were used to compute the mean normalized gray value line profile of the ROIs (Figure 5A,B) to represent the mineral density line profile. The line-profiles of the ROIs were then used to derive the lesion or demineralization depth. The computation of the mean normalized gray value line profile was achieved through the following operations: Surface determination was performed using a native MATLAB edge detection function, and the detected surface was aligned to the level with the highest point of the surface (Figure 5C). Normalized gray value line profile of every line across the surface-aligned ROI was computed and averaged. This was used to derive lesion depth *LD*, which is defined as the distance between the surface and the point where the gray value reached that of sound enamel (Figure 5D). 

### 2.8. Scanning-Electron Microscopy 

Cross-sections of representative samples from each group were scanned using a tabletop environmental SEM (TM-3000, Hitachi, High-Technologies Corporation, Tokyo, Japan) with a BSE detector. The samples were scanned using an accelerating voltage of 15 kV and the Compositional Imaging mode. Prior to scanning, the samples were sectioned using an isomet (Iso-Met^TM^, 11-1180, Buehler Inc., Lake Bluff, IL, USA) inciso-gingivally through the center of the restoration to expose the tooth-restoration interface. Then, they were sonicated in an ultrasonic bath for 5 min. The samples were subsequently polished sequentially with sandpapers of 400, 600, 800 and 1200 grit using a polishing machine (Eco-Met^TM^ 30 variable speed grinder polisher, Buehler, Lake Bluff, IL, USA) with a speed of 50 rpm. A final polish with 1μm aluminum oxide paste was performed. The samples were sonicated again to remove any debris from the polishing. The samples were fixed on an aluminum stub using carbon tapes for scanning.

### 2.9. Primary and Secondary Outcome Variables and Statistical Analysis 

#### 2.9.1. Primary Outcome Variables 

The primary outcome measures compare the effect of “material” on the inhibition of demineralization. The Shapiro–Wilk normality test was firstly used to assess the pattern of data distribution for *LD* and Δ*IR*_50_, Δ*IR*_100_, Δ*IR*_150_, Δ*IR*_200_ and Δ*IR*_250_. All the outcome measures were found to be normally distributed (*p* > 0.05), and therefore, the one-way ANOVA test was used to compare the demineralization inhibitory effects of the four materials. When significant difference was observed, LSD post-hoc tests were then performed. 

#### 2.9.2. Secondary Outcome Variables

The secondary outcome measures compare the effect of “the type of artificial caries model” on the demineralization inhibitory effect of each material. For this, all the outcome measures were found to be normally distributed for the CN, RC and GIC groups, but those for SuO did not. Therefore, in order to compare the outcome measures between the two caries models, the independent-sample *t*-test was used for the CN, RC and GIC groups and the Mann–Whitney test was performed for the SuO group. Additionally, a one-way ANOVA test was used to compare the demineralization inhibitory effects of the four materials in the chemical caries model group.

## 3. Results

### 3.1. Primary Outcome Variables

Comparison of demineralization inhibitory effect of materials (in the biofilm model) on enamel adjacent to the restoration

#### 3.1.1. Lesion Depth (*LD*)

RC demonstrated the deepest mean lesion depth, *D,* of 244.4 ± 54.4 μm, while CN demonstrated the shallowest *D* of 184.2 ± 20.6 μm (Table 2). The mean lesion depth (*D*) for each group is shown in Table 2. The results of the one-way ANOVA test showed that the “material” factor was found to be a significant factor, (*p* = 0.037). A post-hoc LSD test revealed that samples in the CN and GIC groups showed significantly shallower lesion depths (*p* < 0.05) than those in the negative control group, RC, whilst the lesion depth in the SuO group was not significantly different from that of RC (*p* ˃ 0.05). Representative Micro-Ct cross sections from each material and the corresponding OCT scans following demineralization in the biofilm model are illustrated in Figure 6.

#### 3.1.2. Integrated Reflectivity Increase (Δ*IR*)

Concurring with the MicroCt lesion depth results, samples in the RC group showed the highest integrated reflectivity increase (i.e., increase in demineralization severity), while those in the CN group showed the lowest integrated reflectivity increase at all five depths. The results of the one-way ANOVA showed that the means of Δ*IR* for each depth (Δ*IR*_50_, Δ*IR*_100_, Δ*IR*_150_, Δ*IR*_200_ and Δ*IR*_250_) were significantly different (*p* < 0.05) among the four material groups.

Post-hoc LSD tests showed that Δ*IR*_50_, CN and GIC showed significantly lower increases than RC, and the Δ*IR*_50_ of CN were significantly lower than that of GIC with a mean Δ*IR*_50_ of 45.9 ± 38 dB, and 199.9 ± 107 dB, respectively. The Δ*IR*_50_ of SuO, however, were not significantly different from that RC (*p* ˃ 0.05) but were significantly higher than CN and GIC.

For all of the four subsequent depths, RC once again demonstrated a statistically higher (*p* < 0.05) increase in integrated reflectivity than the other three materials. However, contrary to Δ*IR*_50_, the increase in the integrated reflectivity of the subsequent depths for SuO, like that of CN, was not significantly different from that of GIC. In addition to this, CN exhibited a statistically significantly lower increase in integrated reflectivity than SuO (Table 3).

### 3.2. Secondary Outcome Variables

Influence of type of artificial caries model on the demineralization inhibitory effect of each material.

#### 3.2.1. Intra-Material Group Comparison between the Biofilm and Chemical Model

Lesion Depth (*LD*)

The findings from the independent-sample *t*-tests indicated that the “caries model” has a statistically significant effect (*p* < 0.001) on lesion depths for all four of the materials. In the case of RC, the depth of lesions in the biofilm model was 1.9 times greater (Table 2) than in the chemical model, with mean values of 244.4 ± 54 μm and 125 ± 26.1 μm, respectively. A similar pattern was observed in the other three groups, with the lesion depths of the biofilm model being 1.85, 1.7 and 2.02 times greater than that of the chemical model in the SuO, GIC and CN groups, respectively. The mean normalized gray value profiles for each material in both models are presented in Figure 7.

Δ*IR*_50_ (Figure 8)

The Δ*IR*_50_ of the two caries models were not significantly different in both the RC (independent-sample *t*-tests) and SuO groups (Mann–Whitney test because data were not normally distributed). However, Δ*IR*_50_ in the biofilm model of GIC and CN groups (independent-sample *t*-tests) demonstrated statistically significant lesser reflectivity increase (*p* < 0.05) than the chemical model.

Δ*IR*_100_, Δ*IR*_150,_ Δ*IR*_200_ and Δ*IR*_250_ (Figure 8)

The Δ*IR*_100_, Δ*IR*_150_, Δ*IR*_200_ and Δ*IR*_250_ of the two caries models were only significantly different in the RC group, with the biofilm model showing a greater increase than the chemical model. The Δ*IR*_100_, Δ*IR*_150_, Δ*IR*_200_ and Δ*IR*_250_ of the two caries models were not significantly different (*p* ˃ 0.05) in the GIC, CN and SuO groups.

#### 3.2.2. Comparison between Groups in the Chemical Caries Model

(a)Lesion Depth (*LD*)

The one-way ANOVA showed that the lesion depths were not significantly different amongst the four groups.

(b)Integrated Reflectivity Increase (Δ*IR*)

The one-way ANOVA showed that Δ*IR*_50_, Δ*IR*_100_, Δ*IR*_150_, Δ*IR*_200_ and Δ*IR*_250_ were not significantly different amongst the four groups.

### 3.3. Scanning Electron Microscopy Results

An observation of the enamel adjacent to the restoration surface using SEM revealed the presence of a subsurface dark band of demineralization in all of the four materials (Figure 9). In the biofilm model groups, under a magnification of 1.0 K, inter-rod and intra-rod demineralization were seen. In both RC and SuO, a darker band of demineralization was evident, accompanied by the loss of the enamel rods structure in the body of the lesion. CN and GIC showed a lighter band of demineralization, while the rod structure of the enamel could still be identified (Figure 9).

## 4. Discussion

It is well-known that data obtained from clinical studies are the most reliable. However, ethical, economic and time limitations often hinder their application [30]. In vitro chemical and biofilm models have been used as an alternative to simulate carious lesions [6]. In this study, the secondary caries inhibitory effect of ion-releasing restorations was tested in a single-species biofilm model and a chemical pH-cycling model. Bovine incisors were used as a substitute for human teeth, as they were shown to have similar mineral distribution characteristics [31]. The difference between bovine and human enamel was found to be quantitative [32], meaning that the absolute lesion depths might be different between the two, but relative comparisons of intervention efficacy are expected to be similar [33].

The biofilm model is the primary model considered in this study because biofilm models have been found to affect materials’ biodeterioration and ion-exchange [31,32]. To closely mimic in vivo conditions, an in vitro single-species *S. mutans* biofilm model was used in conjunction with the CDC reactor as a form of an artificial mouth model with a continuous flow of sucrose-containing nutrient media. *S. mutans* is one of the main known pathogens in caries formation [34]. It can synthesize water-insoluble glucans from dietary sucrose via a process catalyzed by glucose transferase. Glucans play an important role in adhesive interactions with the tooth surface, as well as in the formation of a cariogenic biofilm matrix that adheres to the tooth surface and prevents the diffusion and interaction of bacterial organic acids with salivary components, which consequentially leads to the decalcification of tooth structure [35]. A single-species biofilm model was used in this study for multiple reasons. Previous research has shown that a *S. mutans*-based caries model was useful to assess the demineralization inhibitory effects of biomaterials [36,37].

In multi-species models, determining the most appropriate species and their relative amounts is challenging. Also, by using a single species model, falsifying influences can be minimized.

The CDC reactor model used in this study was equipped with a pH prob that provided real-time measurement of pH changes within the reactor [38]. A pilot run was conducted first to determine the flow rate that would result in a pH of approximately 4.5. This reactor, when applied to incubate an oral microcosm biofilm, was shown to create caries lesions of approximately 500 μm in depth [39] after 72 h, as determined by cross-polarization optical coherence tomography. The chemical model used in this study is a modified Featherstone pH cycling model for bovine teeth. This model has been validated in a multicenter study, which demonstrated its capability to differentiate between negative and positive controls based on a reduction in ΔZ [26]. The modification made in this study was shortening the duration of the model from 14 to 9 days to be closer to the period of the biofilm model and also modifying the pH to 4.5 for human enamel instead of pH 4.4.

The results indicate that both CN and SuO indeed exhibit demineralization inhibitory effect on the enamel adjacent to it. Our results also showed that the demineralization inhibitory effect is influenced by the type of artificial caries model used. All the samples in the biofilm and chemical models showed carious lesion development that varied in depth between the two models, with more substantial demineralization depth and severity in the biofilm model.

The results of this study show that the biofilm model was able to discriminate the caries inhibitory effect of the ion-releasing composite (CN). Cention N is referred to by the manufacturer as an “alkasite” with the ability to release hydroxide, calcium and fluoride ions from its alkaline calcium-flouro-silicate glass filler [12]. The hydroxide ions present on the surface of the material could have played a role in neutralizing the acid produced by cariogenic bacteria [40]. In addition, the manufacturer postulated that fluoride and calcium ion release may help in preventing demineralization [12]. A previous report showed that CN is capable of continuous hydroxide, fluoride and calcium ion release over a period of 28 days [41], and the concentrations of calcium and fluoride were approximately 300–400 times greater than those of a control of resin composite. These results might explain the shallow lesion depth and lesser demineralization quantified by ΔIR at every depth in comparison to a control of resin composite and the self-adhesive fluoride releasing SuO. In agreement with the results presented in our study, a previous report [42,43] showed that the nanohardness of enamel adjacent to CN was not different from when a conventional GIC was used.

Surefil one (SuO) has been described by the manufacturer as a self-adhesive composite hybrid [10]. It is made up of high molecular weight polyacrylic acid that has been functionalized with polymerizable groups and has a similarity to the polyalkenoate acid copolymer found in Vitremer (3M^TM^) and Ketac Nano (3M^TM^). The fillers are silanized, non-reactive and FAS fillers of various sizes that bind with the resin matrix [10]. The composition indicates that it is an evolution of RMGIC. This partially water-based material, in theory, promotes water and ion exchange with the oral environment. This results in the release of fluoride, aluminum and calcium ions as well as possibly other ions due to the composition of the reactive fillers. The findings of this study suggest that SuO demonstrates the capability to inhibit demineralization near the restoration border. The OCT results indicate a reduction in the severity of demineralization in SuO compared to the bulk-fill resin composite Powerfill. However, when examining lesion depth using Micro-CT, these results do not support the aforementioned observation. This inconsistency can be attributed to the discrepancy in the region of interest (ROI) width between OCT (0.25 mm) and Micro-CT (1.5 mm). Considering that SuO is an evolution of RMGIC, it is expected that its ability to prevent caries is primarily limited to the area adjacent to the restoration margin. Previous studies have also reported the capacity of various GIC types, including RMGIC, to decrease demineralization near restoration margins [37,44].

The reasons for the difference in results observed between the two models may be due to several factors. One of them might related to the biofilm that acts as a reservoir or diffusion barrier for demineralization-inhibiting ions that diffused out from the restorative material [45]. Biofilms may act as a reservoir to slow the sustained influx of acid into enamel as well as the outflux of calcium and phosphate released from the enamel surfaces, thereby protecting the underlying demineralized enamel [45].

It was previously reported that the calcium binding capacity of *S. mutans* is 30 μmol/g in wet weight [46]. However, in a study by Zhang et al. [45], the calcium concentration in the biofilm treated by nano-hydroxyapatite (nHA) was twice that value. According to the authors, this could be attributed to the large surface area of nHA, which favored their retention in the biofilm thus inhibiting demineralization. Depending on the type of the anti-caries agent, the presence of biofilm might decrease or increase demineralization. In fluoride-releasing materials, the presence of biofilm might inhibit the remineralizing potential of fluoride. It was reported that the diffusion coefficient of fluoride in an in situ incubated biofilm was 40% of that in water [47] due to the biofilm structure and the biofilm thickness; consequently, the fluoride level at the biofilm-enamel interface might have been too low to achieve net remineralization. The caries inhibitory effect of ion-releasing restorations could not be distinguished by the pH cycling model due to the absence of a glucan matrix that promotes adhesion to the tooth or composite surface. Consequently, the enamel surface lacking the presence of the trapped glucan matrix exhibits greater resistance to acid attacks, making it easier to remineralize and resulting in the reduced elution of ions [25]. The results of this study are in agreement with those of Ionescu et al.’s work [30], where an open cycle bioreactor model was able to produce higher demineralization depths while differentiating the demineralization inhibitory effect of fluoride-releasing (RMGIC) from that of resin composite.

Although the two ion-releasing restorations examined in this research demonstrated a demineralization inhibitory effect comparable to GIC (glass ionomer cement), further investigation is necessary to assess the impact of ion release on mechanical properties and wear [48].

Optical coherence tomography has been successfully used as a non-invasive method capable of detecting and quantifying the early carious lesions [49]. The most straightforward way to assess the severity of a lesion using OCT is to measure the reflectivity of the light from the lesion area over its depth. However, this method has limitations because the strong reflectivity of the tooth surface can overshadow the signal from the lesion, making it difficult to accurately quantify the severity of the lesion [49]. In order to avoid the high surface reflection, the integration of the A-scan in this study did not include the first 30 μ at the surface. Measuring integrated reflectivity at different depths helped in detecting lesion severity between the two models.

This work has a limitation that relates to the use of single-species biofilm model, which may not be able to represent the complexity of how secondary caries develop in the oral environment and may give an exaggerated estimate of the cariogenicity of *S. mutans*. Although *S. mutans* is an important contributor to secondary caries development, it has interactions with other microbial species that are not considered in single-species biofilm models. Furthermore, the simplification of in vitro studies when attempting to replicate complex clinical situations was inevitable, which emphasizes the need for the standardization and calibration of in vitro caries models.

Moreover, although OCT has been confirmed as a reliable technique for measuring enamel demineralization, it is crucial to exercise caution when interpreting the results obtained from integrated reflectivity measurements. This is due to various factors, including surface roughness, the presence of air bubbles and the limited depth of penetration of OCT, all of which can impact the outcomes and must be considered.

## 5. Conclusions

We conclude that the alkasite Cention N and the self-adhesive composite hybrid Surefil one both exhibit demineralization inhibitory effect on enamel adjacent to the tooth-restoration margin, similar to that of GIC. However, the demineralization inhibitory effect of Cention N is more pronounced than that of Surefil, both in depth and in the degree of demineralization. We can conclude that the degree of the inhibitory effect of ion-releasing restorations is different in a biofilm than that in a chemical caries model. The biofilm model used in this study was able to discriminate the demineralization inhibitory effect of the ion-releasing restorations more than the chemical model.

## Figures and Tables

**Figure 1 materials-16-05558-f001:**
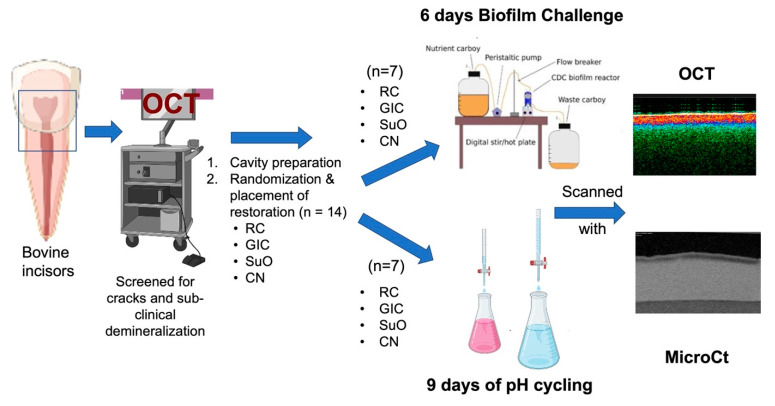
Schematic illustration of the study flow.

**Figure 2 materials-16-05558-f002:**
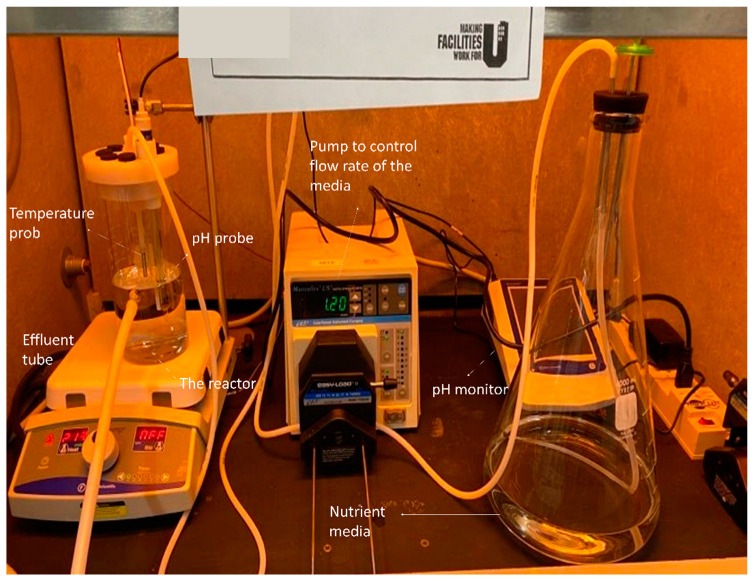
Set-up of the CDC reactor.

**Figure 3 materials-16-05558-f003:**
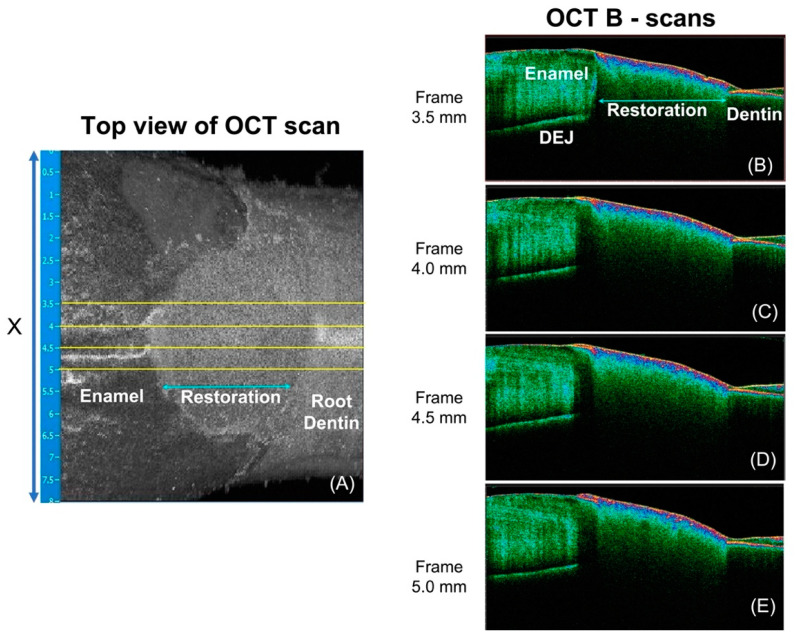
This figure shows an OCT top view of a sample as it appears on the Santec *Innervision* OCT capturing software. (**A**) shows the location of the 4 interspersed cross-sections (solid lines) chosen for computation of mean reflectance depth profile. (**B**–**E**) are the corresponding chosen cross-sections. The positions of the enamel, restoration, dentin and enamel-dentine junction (EDJ) are labelled in (**A**,**B**).

**Figure 4 materials-16-05558-f004:**
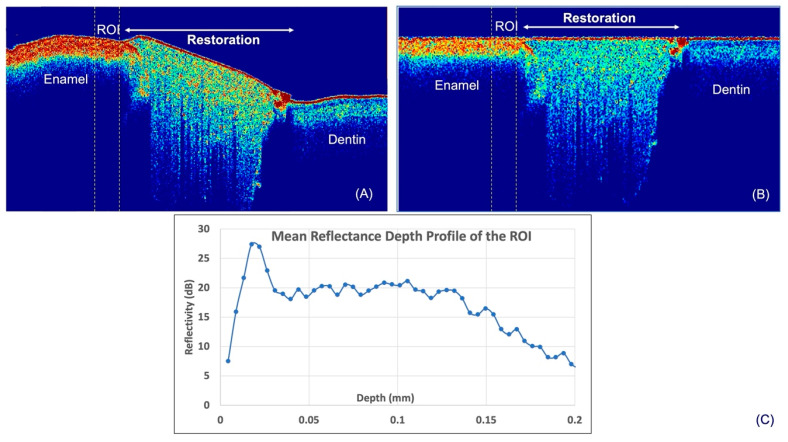
Steps for computation of the mean reflectance depth profile of the region of interest. (**A**) shows the selection of ROI before surface alignment, (**B**) the OCT image after alignment, (**C**) mean A-scan of ROI.

**Figure 5 materials-16-05558-f005:**
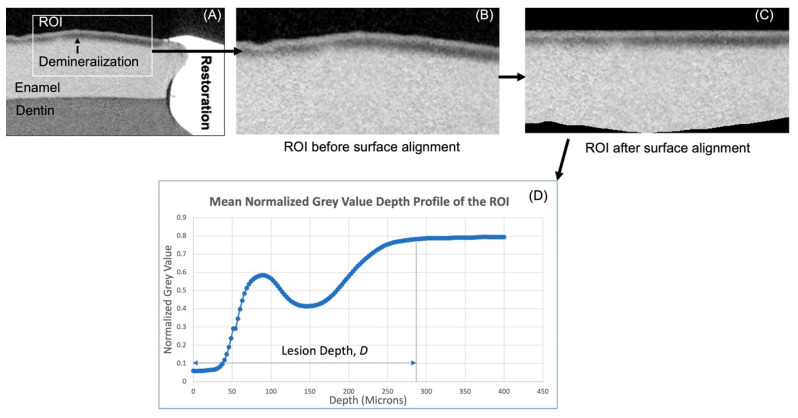
Steps for computation of the mean gray value density line profile of the region of interest. (**A**) showing cross section as it appears on VGSTUDIO, (**B**) ROI before alignment, (**C**) ROI after alignment and (**D**) mean normalized depth-gray value profile.

**Figure 6 materials-16-05558-f006:**
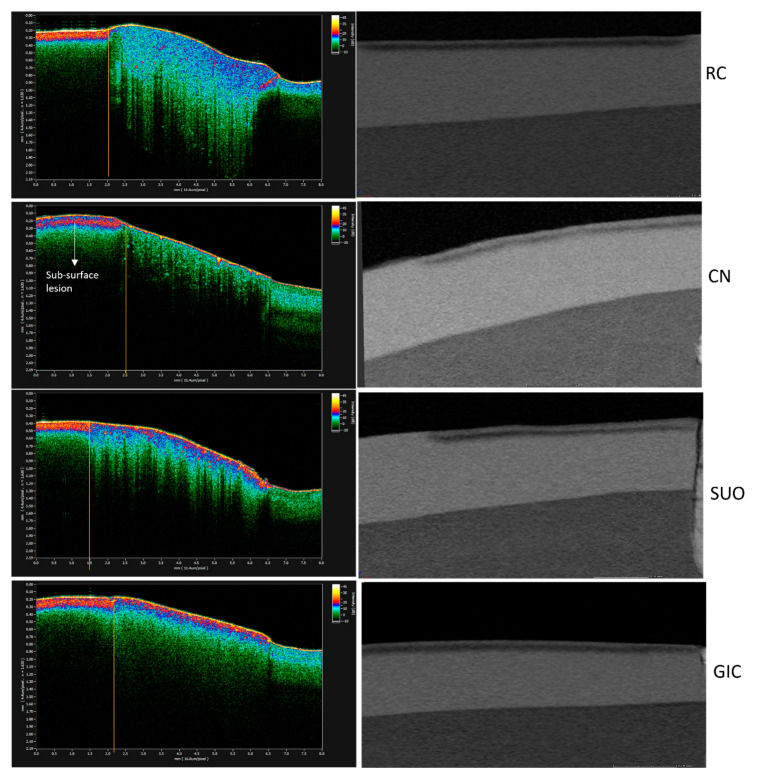
Representative OCT B-scans and the corresponding Micro-Ct cross sections of each group after biofilm demineralization. Note that the lesion in the CN group was less homogenous than the other groups and the hyper-mineralized surface layer is more prominent than the other groups.

**Figure 7 materials-16-05558-f007:**
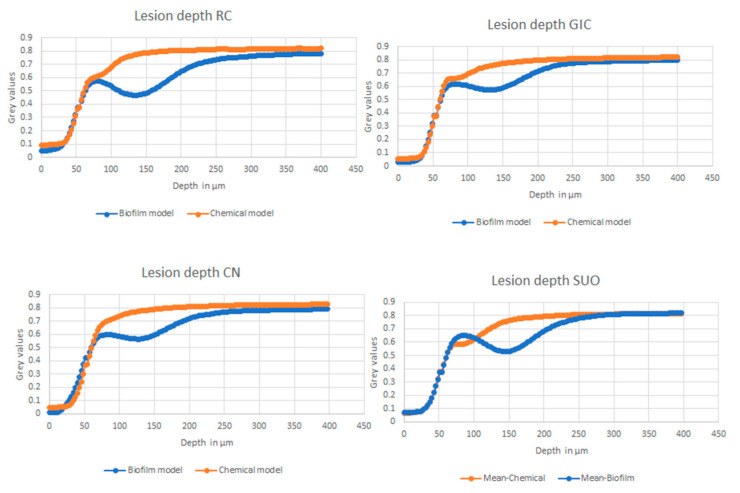
Mean normalized grey value profiles for each material, showing lesion depth (LD).

**Figure 8 materials-16-05558-f008:**
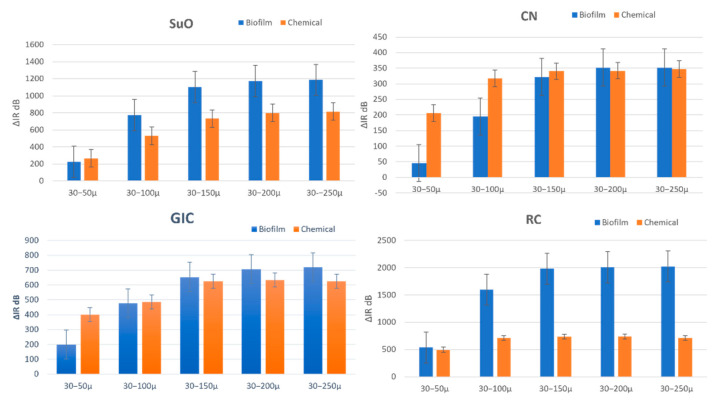
Showing Δ*IR*_50_, Δ*IR*_100_, Δ*IR*_150_, Δ*IR*_200_ and Δ*IR*_250_ for each material in both models.

**Figure 9 materials-16-05558-f009:**
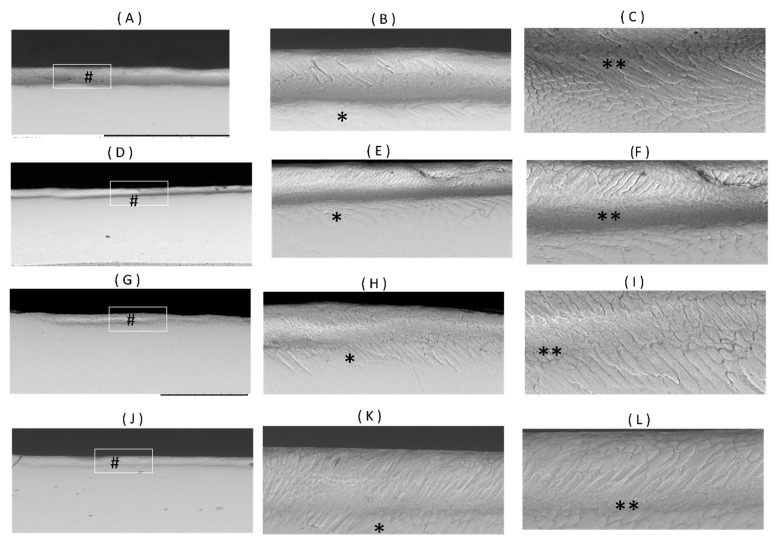
SEM images of a representative sample in each group after demineralization showing (#) subsurface band of demineralization, (*) areas of inter-rod demineralization and (**) areas of intra-rod demineralization. Figures (**A**–**C**) are from the RC_biofilm group at 200×, 600× and 1000× magnifications; (**D**–**F**) are from the SuO_biofilm group at 120×, 500× and 1000× magnifications; (**G**–**I**) are from the CN_biofilm group at 120×, 500× and 1000× magnifications; (**J**–**L**) are from the GIC_biofilm group at 120×, 500× and 1000× magnifications.

**Table 1 materials-16-05558-t001:** Materials used in this study.

Material	Classification (According to Manufacturer)	Manufacturer	Main Composition	Lot Number
Surefil one	Self-adhesive, Bulk-fill composite, Hybrid	Dentsply Sirona, Charlotte, NC, USA	Modified polyacid (MOPOS), Bifunctional acrylate (BADEP), Acrylic acid, Water, Reactive and non-reactive glass filler, Initiator, Stabilizer	2107000263
Cention N	Alkasite, Ion releasing self-cure composite	Ivoclar Vivadent AG, Schaan, Liechtenstein	Liquid: (% weight) Dimethacrylate (95–97%), Additive (1–2%), Initiator (2–3%), Stabilizer (<1%). Powder: Calcium fluorosilicate glass (25–35%), Ba-Al silicate glass (20–30%), Ca-Ba-Al fluorosilicate glass (10–20%), Ytterbium trifluoride (5–10%)	V26429
Ketac Molar	GIC	3m ESPE, GmbH, Germany	Powder: aluminum-calcium-lanthanum-fluorosilicate glass, 5% spray dried ESPE polycarbonate acid (which is a copolymer of acrylic and maleic acid) Liquid: polycarbonic acid and tartaric acid	6405794
Tetric Powerfill	Bulk-fill resin composite	Ivoclar Vivadent AG, Schaan, Liechtenstein	Dimethacrylates (19.7%), Prepolymer (17%), Barium glass filler, Ytterbium trifluoride, Mixed oxide (62.5%), Additive, Initiators, stabilizers Pigments (<1.0%)	Z01NZP
Adhese Universal	Universal self-etch adhesive	Ivoclar Vivadent AG, Schaan, Liechtenstein	Methacrylates (60–70%), Water, Ethanol (23–28%) Highly dispersed silicon dioxide (3–5%), Initiators and stabilizers (3–5%)	ZL0B69

**Table 2 materials-16-05558-t002:** Mean lesion depth, *LD*, for each group and multiple comparisons between groups.

	Biofilm		Chemical		Ratio (Biofilm/Chemical)
	Mean (μm)	SD (μm)	Mean (μm)	SD (μm)	
RC	244.4 ^A^	54.4	125 ^a^	26.1	1.9
CN	184.2 ^B^	20.6	91.1 ^a^	21.5	2.02
GIC	197.5 ^B^	37.8	116.1 ^a^	34.3	1.7
SuO	221.8 ^AB^	34.1	119.8 ^a^	22.5	1.85

In the biofilm model, means with the same superscripted uppercase letters are not statistically significant (*p* > 0.05). In the chemical model, means with the same superscripted lowercase letters are not statistically significant (*p* > 0.05).

**Table 3 materials-16-05558-t003:** Means ± SD of Δ*IR*_50_, Δ*IR*_100_, Δ*IR*_150_, Δ*IR*_200_ and Δ*IR*_250_ for ROI_1_ of the biofilm model and the results of one-way ANOVA and multiple comparisons (LSD post hoc) between groups.

	Δ*IR*_50_ (dB)		Δ*IR*_100_ (dB)		Δ*IR*_150_ (dB)		Δ*IR*_200_ (dB)		Δ*IR*_250_ (dB)	
Material	Mean ± SD	*p* Value	Mean ± SD	*p* Value	Mean ± SD	*p* Value	Mean ± SD	*p* Value	Mean ± SD	*p* Value
SuO	226 ± 114 ^A^	0.037	774.7 ± 544.8 ^B^	<0.001	1105.1 ± 671 ^B^	<0.001	1174.2 ± 652 ^B^	<0.001	1187.6 ± 631 ^B^	<0.001
CN	45.9 ± 38 ^B^		195.7 ± 65.3 ^C^		322.3 ± 95.9 ^C^		352.7 ± 278 ^C^		352.5 ± 109.7 ^C^	
GIC	199.9 ± 107 ^B^		477.4 ± 276 ^BC^		654.8 ± 298 ^BC^		707.7 ± 300.5 ^BC^		719.3 ± 299 ^BC^	
RC	535 ± 185 ^A^		1599.9 ± 798 ^A^		1938.7 ± 817 ^A^		2011.1 ± 793 ^A^		2026.1 ± 770 ^A^	

Footnote: For each variable, groups with the same superscripted letters are not statistically different from each other.

## Data Availability

The data presented here are available on request from the corresponding author.

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
