# Peer review of "Inhibition of Caries around Restoration by Ion-Releasing Restorative Materials: An In Vitro Optical Coherence Tomography and Micro-Computed Tomography Evaluation"

_materials, 2023, doi:10.3390/ma16165558_

Round 1

Reviewer 1 Report

“Inhibition of Caries Around Restoration by Ion-Releasing Restorative Materials: An In vitro Optical Coherence Tomography (OCT) And Micro-Ct Evaluation” was submitted to Materials.

The manuscript deals with an interesting issuehoweverthere are several concerns related to the study

Title: please delete OCT and define -Ct

Abstract

-Line 18. Please define OCT and -Ct.

-Line 19. Please replace D to LD. Please consider this through the manuscript.

-Line 23. “Statistically deeper”. Please present the p-value.

Keywords: Terms should be reviewed. Some are not MeSH terms.

Introduction

- Lines 33,34. Please add references.

- Lines 47-50. Please add references.

- Line 70. References 11, 12 are very old. Please present some with good level of evidence.

- Line 84. “Expensive than clinical trials”. However, in vitro studies have a lower level of evidence. Please comment on this.

Methods

- Please highlight the primary and secondary outcome variables.

- No information related to the operators who performed the procedures is presented. The number, the calibration process and the results of that calibration must be presented along with the statistical test used.

- Lines 117-118. The information obtained from reference 25 must be detailed. Not only alpha and beta are essential for calculating the sample size.

- Lines 141 and 191 describe randomization process. Please detail them.

- Line 207. Please define CDC.

- Line 219. Please revise the form to write microorganisms scientifically. 

- Line 219. Please define MSSB.

- Please improve figure 2. Some legends are incomplete.

- Line 287. Please define EDJ.

- Line 339. Please present the p-value of the normality test.

Results

- Please present the results according to primary and secondary outcome variables.

- Please improve figure 9. It is difficult to read the very small legends.

- Lines 358-359. The information related to the normal distribution of the data has already been presented.

Discussion

-Line 447 please add references.

-Line 459. Reference 33 is very old.

-An in vitro design is also a limitation.

-Please revise some references. References 11 to 16, 22, 24, 33, and 44 are very old.

Minor

Author Response

Please  refer to the file attached.

Reviewer 2 Report

In recent years there has been a notable increase in the prevalence of root caries

The objectives of this in vitro study are Compare the demineralization inhibitory effect of two ion-releasing restorative materials to that of a conventional resin composite and a conventional glass ionomer material. And 2. Evaluate the influence of the type of artificial caries model on the demineralization inhibitory effect of the four materials mentioned above.

For this reason, the authors need to show the subject area compared with other published material.

the manuscript is interesting, my recommendation is to make the most detailed and short introduction on the subject investigated to clarify the purpose of the study.

The manuscripts should be published because of the results they obtain.

A missing conclusion is to analyze the information obtained in comparison with the literature.

Improve the images. They conclude for example that “the degree of inhibitory effect of ion releasing restorations is different in a biofilm than that in a chemical caries model” Why?.

In recent years there has been a notable increase in the prevalence of root caries

The objectives of this in vitro study are Compare the demineralization inhibitory effect of two ion-releasing restorative materials to that of a conventional resin composite and a conventional glass ionomer material. And 2. Evaluate the influence of the type of artificial caries model on the demineralization inhibitory effect of the four materials mentioned above.

For this reason, the authors need to show the subject area compared with other published material.

the manuscript is interesting, my recommendation is to make the most detailed and short introduction on the subject investigated to clarify the purpose of the study.

The manuscripts should be published because of the results they obtain.

A missing conclusion is to analyze the information obtained in comparison with the literature.

Improve the images. They conclude for example that “the degree of inhibitory effect of ion releasing restorations is different in a biofilm than that in a chemical caries model” Why?.

Author Response

Please refer to the file attached.

Reviewer 3 Report

INTRODUCTION

“…With resin composite approaching 50 years of clinical service…”- Please replace "service" with the word use or utilization.

Lines 34, 37, 39, 50, 51, 65, 83 – the references are missing.

The authors must have presented a null hypothesis.

I suggest that the authors should read and refer the article:

10.3390/app11178256

MATERIALS AND METHODS

Why were bovine teeth and not human teeth (tooth bank) chosen?

“…The teeth were stored in 0.1% thymol until used…” - Where were they stored and at what temperature? During 3 months the storage liquid was changed? how many times?

4 groups (n=14) - How did the authors find n=14 to be ideal?

What were the control group(s)? Did the authors not consider them?

Why did the authors use this adhesive system? They chose a universal that is not the one with the highest adhesive values. Did the authors apply only one layer of adhesive? Why only one coat?...

We do not know at what temperature and where the restored specimens were stored.

Didn't the authors consider the possibility of also incubating with lactobacillus, thus enhancing the development of the carious lesion, as it happens in humans?

DISCUSSION

It is possible to do this study with human teeth. Authors should discuss the differences between the use of bovine teeth and human teeth - this is an in vitro study and not a clinical trial or an in vivo study.

There is no control group, which is important for analyzing the results and, consequently, for carrying out the discussion.

Round 2

Reviewer 1 Report

The authors improved the manuscript 

Minor editing 

Reviewer 3 Report

With the changes proposed by the reviewer and accepted by the authors, the article could be considered accepted for publication.